# Polyindole-Derived Nitrogen-Doped Graphene Quantum Dots-Based Electrochemical Sensor for Dopamine Detection

**DOI:** 10.3390/bios12121063

**Published:** 2022-11-22

**Authors:** Anjitha Thadathil, Dipin Thacharakkal, Yahya A. Ismail, Pradeepan Periyat

**Affiliations:** 1Department of Chemistry, University of Calicut, Malappuram 673635, India; 2Department of Environmental Studies, Kannur University, Kannur 670567, India

**Keywords:** graphene quantum dots, dopamine, sensitivity, selectivity

## Abstract

The sensitive monitoring of dopamine levels in the human body is of utmost importance since its abnormal levels can cause a variety of medical and behavioral problems. In this regard, we report the synthesis of nitrogen-doped graphene quantum dots (N-GQDs) from polyindole (PIN) via a facile single-step hydrothermal synthetic strategy that can act as an efficient electrochemical catalyst for the detection of dopamine (DA). The average diameter of N-GQDs was ∼5.2 nm and showed a C/N atomic ratio of ∼2.75%. These N-GQDs exhibit a cyan fluorescence color under irradiation from a 365 nm lamp, while PIN has no characteristic PL. The presence of richly N-doped graphitic lattices in the N-GQDs possibly accounts for the improved catalytic activity of N-GQDs/GCE towards electrocatalytic DA detection. Under optimum conditions, this novel N-GQDs-modified electrode exhibits superior selectivity and sensitivity. Moreover, it could detect as low as 0.15 nM of DA with a linear range of 0.001–1000 µM. In addition, the outstanding sensing attributes of the detector were extended to the real samples as well. Overall, our findings evidence that N-GQDs-based DA electrochemical sensors can be synthesized from PIN precursor and could act as promising EC sensors in medical diagnostic applications.

## 1. Introduction

Zero-dimensional graphene quantum dots (GQDs) can be described as shining stars among the luminescent carbon nanoparticles, which own graphene lattices within the dots irrespective of the dot size [1,2]. GQDs consist of graphene sheets with a diameter smaller than 20 nm and display exciting novel properties such as high photostability, good biocompatibility, resistance against photobleaching, tunable photoluminescence, low cytotoxicity, slow hot-carrier relaxation, good electrical conductivity, and good dispersion in various solvents, due to the edge effect and the quantum confinement effect (QCE) [3,4,5,6,7]. Since GQDs are non-metallic fluorophores, they own the benefits of low cost and wide availability, biocompatibility, simple preparation strategies, easy operation, less toxic effects, and improved surface functionalization characteristics, making them desirable candidates for application in bioimaging, biolabeling, drug delivery, photovoltaic devices, biosensors, light-emitting diodes, and fuel cells [8,9,10,11,12]. As research has developed, it was found that doping with heteroatoms such as nitrogen (N) can efficiently modulate the local chemical features and its band gaps, and offer more active sites, tunable electronic characteristics, and optical properties for electrochemical sensing applications [13,14,15]. For instance, Saisree et al. reported radical sensitivity and selectivity in the electrochemical sensing of cadmium ions in water by polyaniline-derived nitrogen-doped graphene quantum dots [16]. Y. Liu et al. fabricated a label-free photoelectrochemical aptasensor based on nitrogen-doped graphene quantum dots for chloramphenicol determination [17]. Li and co-workers reported new N-doped graphene quantum dots-decorated N-doped carbon nanofiber (NGQDs@NCNFs) composites for nitrile detection [18].

To date, various bottom-up and top-down methods were employed for the synthesis of N-doped GQDs (N-GQDs). The top-down method usually involves the cutting down of carbon nanomaterial in the presence of N-precursors [19,20,21,22,23,24]. This method comprises multiple steps, loss of control over the size of products, heterogeneous morphology, less production, and quantum yields of GQDs limit bulk production [2]. Additionally, this would create many defects in the aromatic structure, and the percentage of N-doping within the lattice will be lower. The bottom-up method includes the synthesis of quantum dots from glucose, cyclodextrin, citric acid, and so forth, in the presence of N-precursors [25,26,27,28,29,30]. Indeed, the simplicity in operation and ambient reaction conditions offered by the hydrothermal approach is preferred over the other methods for the formation of N-GQDs with N-rich aromatic domains [1]. The conducting polymers with N-heteroatoms were expected to undergo ring closure in a hydrothermal process, which results in the formation of richly N-doped N-GQDs [31]. 

Dopamine (DA), often called the happy hormone, or feel-good hormone is essential for different neuronal activities such as memory, learning, attention, cognition, behavior, movement, and emotion [32]. The abnormalities in dopamine levels in the human body may lead to several health problems including Parkinson’s disease, schizophrenia, epilepsy, and senile dementia [33,34]. Therefore, enormous efforts are employed throughout the last three decades for the selective and sensitive detection of dopamine using techniques such as capillary electrophoresis, liquid chromatography–electrospray tandem mass spectrometry, high-performance liquid chromatography, fluorescence spectrometry, UV–visible spectrophotometry, and enzymatic methods [35,36,37,38,39]. Still, these techniques usually involve time-consuming and complicated pretreatment procedures, costly materials, and require expensive instruments, thus greatly hindering their widespread practical applications [40]. In this context, electrochemical sensors have been used as a potential analytical method to detect ultra-sensitive concentrations of various analytes in a specific and sensitive route [41,42,43,44,45]. Since dopamine exhibits significant electroactivity, its sensing has been widely studied by several electrochemical (EC) methods. The salient features of this method include relatively low cost, simplicity, accuracy, portability of equipment, ultra-high sensitivity, and selectivity [46,47,48]. However, the main difficulty arises due to the interference of ascorbic acid (AA) and uric acid (UA). Dopamine always coexists with UA and AA in biological fluids, with similar oxidation potentials [49]. Further, the oxidation products of these interfering species might be deposited on the surface of the electrode and badly affect the reproducibility and selectivity of the sensor, resulting in its fouling [50]. Studies show that sensing electrode materials such as graphene, carbon nanotubes, gold nanoparticles such as polyaniline, and others can tackle this problem [51,52,53]. Regardless of the selective sensing of dopamine using these materials, other concerns are the difficult synthesis protocol, sensitivity, and high cost. Thus, developing a sensor that is not only selective, sensitive, and reliable, but also practical and economical is still challenging.

Sami Ben Aoun fabricated a nanostructured carbon electrode-modified N-GQDs–chitosan hybrid-based electrochemical sensor determination of DA within the linear ranges 1–100 µM and 100–200 µM [54]. Unfortunately, in many of the previous reports of EC determination, the linear range of sensors cannot cover the physiological range (0.01–1 μM) and displays a higher limit of detection (>0.01 μM) [40]. 

Polyindole (PIN) is a rising conducting polymer comprising a fused aromatic structure with a benzene ring and a pyrrole ring [55,56,57,58]. The synthesis of the PIN can be accomplished via the facile in situ emulsion polymerization method at room temperature and could act as an excellent N-containing aromatic carbon precursor for the preparation of N-GQDs [59,60]. In our work, we report a simple one-step fool-proof method for synthesizing N-doped GQDs, using hydrothermal reduction of polyindole (PIN). The presence of pyrrolic, pyridinic, and graphitic N rings arising via the in situ N-doping was expected to enhance the electronic properties and aid in the sensitive and selective electrochemical determination of dopamine. 

In the present work, we report a single-step fool-proof method for synthesizing N-GQDs from polyindole (PIN) via hydrothermal reduction, and electrochemical sensing of DA using these N-GQDs. This N-GQDs-based EC sensor could detect as low as 0.15 nM of DA with a linear range of 0.001–1000 µM. Moreover, the sensor was able to distinguish between AA, UA, and DA and demonstrated outstanding stability properties, reproducibility, and applicability in real human samples as well.

## 2. Materials and Methods

### 2.1. Chemicals

Indole powder, sodium hydroxide (NaOH), and sodium dodecylbenzene sulfonate (SDBS) were procured from Himedia, India Ltd. (Mumbai, India). Pb(NO_3_)_2,_ MgCl_2_, KCl, NaCl, Ca(NO_3_)_2_, Na_2_HPO_4_, NaH_2_PO_4_, MnCl_2_, and FeCl_3_ were purchased from Merck Chemical Co., Ltd. (Mumbai, India). Ascorbic acid, dopamine hydrochloride, citric acid, glucose, sucrose, alanine, glycine, phenylalanine, valine, urea, and uric acid were procured from Sigma Aldrich, India Ltd. (Mumbai, India). An appropriate amount of Na_2_HPO_4_ and NaH_2_PO_4_ was used to prepare phosphate-buffered saline (PBS) of different pH by adjusting with 100 mM NaOH and H_3_PO_4_.

### 2.2. Synthesis of PIN

The synthesis of PIN was performed via in situ emulsion polymerization of indole monomer. For typical synthesis, 0.298 g of SDBS in 10 mL of distilled water was mixed with 3 g indole in 50 mL of the ethanol–water mixture (1:1 *v*/*v*). After 30 min of sonication, 1.75 g of FeCl_3_ in 10 mL water was added dropwise to the above solution until the solution showed a dark green color. Magnetic stirring was continued for 24 h to complete the polymerization reaction. Finally, the dark green-colored precipitate acquired was washed with ethanol and distilled water multiple times and dried in an oven at 70 °C to obtain PIN nanopowders.

### 2.3. Synthesis of N-GQDs

A facile one-step hydrothermal reduction method was employed in the synthesis of N-GQDs from PIN precursor material [31] (Appendix A). For typical synthesis, 300 mg of as-synthesized PIN powder was ultrasonically dispersed in 60 mL of distilled water for 1 h. Subsequently, the PIN dispersion was mixed with 1 mL of NaOH (2 M). The resultant solution was then transferred to a Teflon-lined autoclave and kept at 220 °C for 12 h. Afterward, the solution in the autoclave was allowed to cool naturally to room temperature. Finally, the unwanted dark brown-colored PIN powers settled at the bottom were removed from the solution via ultracentrifugation, repeated 3 times at 4000 rpm and for 5 min. This supernatant yellow-colored solution containing N-GQDs was diluted 2 fold for electrochemical sensing characterizations.

### 2.4. Characterizations

The chemical structures of the PIN and N-GQDs were studied by employing Fourier transform infrared spectrophotometer, Model 4100 (JASCO, Seoul, Republic of Korea) via the KBr pellet method. The size distribution and morphology of N-GQDs were measured using a high-resolution transmission electron microscope (HR-TEM) JEM2100 (Jeol Ltd., Tokyo, Japan). A UV–vis NIR spectrophotometer model V-550 (JASCO, Porland, OR, USA) was used for UV–vis spectra analysis. An X’pert3 Powdered X-ray diffractometer (Malvern Panalytical, Malvern, UK) with CuK_α_ (λ = 1.5406 Å) radiation was used for the phase identification characterizations. An X-ray photoelectron spectrometer (Omicron Nanotechnology Ltd., Taunusstein, Germany) was employed for XPS analyses. A laser micro-Raman spectrometer model NRS-4100 (JASCO, Seoul, Republic of Korea) was used for Raman spectral analysis. PL spectra studies were conducted on a FluoroMax-4 (Horiba, Longjumeau, France) instrument.

### 2.5. Fabrication of Electrodes for EC Sensing Application

Before use, glassy carbon electrodes (GCE, 3 mm in diameter) were sequentially polished using a wetted microcloth to produce a mirror finish and followed by 2 min ultra-sonication in distilled water and acetone. The N-GQDs-modified electrode was fabricated by drop-casting 10 μL of an aqueous suspension of N-GQDs on a glassy carbon electrode (GCE) using a micropipette and dried at ambient temperature (N-GQDs/GCE). Likewise, the GC-modified PIN electrode was also fabricated for comparison (PIN/GCE). The Zahner Zennium Pro electrochemical workstation (Chennai, India), connected to a personal computer with Thales XT electrochemical software, was employed to perform electrochemical measurements. The three-electrode system comprising a modified GCE (working electrode), Ag(s)/AgCl(s)/Cl^1−^ (aq.) (reference electrode) and a platinum wire (counter electrode) was used in the electrochemical characterization. The cyclic voltammetry (CV), chronoamperometry, and linear sweep voltammetry (LSV) measurements were used to study the EC responses of N-GQDs/GCE, GCE, and PIN/GCE in PBS, 0.1 M at RT.

### 2.6. Sample Preparation

Urine samples were collected from healthy volunteers, and kindly provided by a health care laboratory (Malappuram, India). All the collected urine samples were centrifuged for 5 min at 7000 rpm and diluted (100 fold) with PBS solutions (pH 7.0) without any other pretreatment before analysis. The test solution was prepared by spiking the filtrate with the known amount of DA.

## 3. Results

### 3.1. The Physiochemical Characterization of N-GQDs

The formation of N-GQDs was first confirmed from the TEM images (Figure 1). The size distribution curve shown in the inset of Figure 1a depicts the size distribution of N-GQDs within the range of 0.2–20 nm. The average diameter of N-GQDs was ∼5.2 nm. The HR-TEM image shows an apparently round shape for the N-GQDs with visible crystalline lattice fringes (Figure 1b). The inter-planar spacing measured from the HR-TEM image was 0.21 nm, corresponding to sp^2^ graphitic diffraction planes (002) [61]. The selected-area electron diffraction (SAED) pattern of a single N-GQDs was taken [inset Figure 1b], and the rings on the diffraction pattern depict a crystal structure of the N-GQDs. Further, the appearance of the arc in the SAED pattern corresponds to the XRD data of the N-GQDs. Figure 1c illustrates the XPS survey scan spectrum for the N-GQDs, revealing C 1s peak at ∼289.6 eV, N 1s peak at 400.5 eV, and O 1s peak at ∼532.6 eV. After integrating the peak area, the C/O atomic ratio was estimated (∼2.41%), which is slightly greater than that of nanometer-sized GO reported in the literature (∼2%) [62]. Meanwhile, the estimated C/N atomic ratio is ∼2.75%, which is comparable with that of the N-GQDs synthesized by other methods and confirms the N-doping in graphitic lattices [63,64]. The C 1s peak is deconvoluted by a dominant C-C bond (284.7 eV), and C=O (285.9 eV), C=N (285.19 eV), and O-C=O (288.6 eV) bonds (Figure 1d). The presence of the C=N peak (∼285.19 eV) clearly demonstrates the N-doping via the hydrothermal reduction of PIN. The N 1s peak is deconvoluted to pyrrolic (∼399.0 eV), pyridinic (∼398.2 eV), and graphitic (∼399.64 eV) nitrogen peaks, which portrays the in situ N-doping in the graphitic lattice (Figure 1e). Further, the surface atomic compositions of pyrrolic and pyridinic nitrogen were estimated as 33.92 at.% and ∼10.29 at.%, respectively. In agreement with the proposed structure of N-GQD given in Appendix A, a higher proportion of graphitic-N was estimated (55.78 at.%) from Figure 1e. Additionally, the O 1s spectrum (Figure 1f) deconvoluted two distinctive oxygen states of C– O–C/C–OH (531.82 eV) and C=O (530.90 eV). The XPS analysis of PIN is also provided for better comparison (Appendix A).

Figure 2a shows the FTIR spectra of N-GQD and PIN in the range of 4000–400 cm^−1^. The characteristic IR bands of PIN were found at 3405, 3116, 3043, 1584, 1622, 1456, 1343, 1105 and 747 cm^−1^ [65]. 

Figure 2a shows the FTIR spectra of N-GQD and PIN in the range of 4000–400 cm^−1^. The characteristic IR bands of PIN were found at 3405, 3116, 3043, 1584, 1622, 1456, 1343, 1105 and 747 cm^−1^ [65]. The broad band that occurred at 3405 cm^−1^ corresponds to the N-H starching vibrations of PIN. The bands at 3116 and 3043 cm^−1^ are ascribed to the characteristic C-H stretching vibrations in the PIN. The band at 1584 cm^−1^ is due to the N-H deformation vibrations in the PIN. The absorptions at 1622 and 1456 cm^−1^ are assigned to C=C stretching, and C-C stretching, respectively. The band at 1343 cm^−1^ is due to the characteristic C-N stretching vibration, whereas the band at 1,105,747 cm^−1^ relates to the C-H deformations in the PIN. In N-GQD, a peak at 3440 cm^−1^ denotes the O-H/ N-H stretching vibrations. Further, the C-N in-plane bending vibration at 1000 cm^−1^ along with C-NH-C stretching at 1380 cm^−1^ confirmed the presence of N-doped N-GQDs [31]. The presence of –C-H stretching of an aromatic ring is observed at approximately 2918, and 2849 cm^−1^, respectively [1]. In addition, the band that occurred at 1533 cm^−1^ is ascribed to sp^2^-bonded C-N. Moreover, the prominent absorptions at 1628 (C=C stretching), 1737 (C=O stretching), 1450 (-COO stretching), and 1120 cm^−1^ (–C-OH stretching) agree with the presence of the aromatic carbon lattice within N-GQD along with hydrophilic edge functionalities. In summary, the FTIR spectra revealed that the aromatic ring structure of PIN was retained within N-GQDS.

The zeta potential measurement of N-GQDs shown in Figure 2c strongly suggests the presence of carbonyl groups on the surface, which makes N-GQDs highly water soluble. In the XRD pattern (Figure 2c), N-GQDs show a peak at 2θ = 23.5° referring to the reflections of graphitic planes (002) [66]. Furthermore, the presence of disordered sp^2^ graphitic lattices is evidenced from the Raman spectrum of the N-GQDs (Figure 2d) [67]. Upon deconvolution, the Raman spectra display the D band and G band at 1317 and 1768 cm^−1^, respectively. In addition, a 2D band was identified at 3015 cm^−1^ [68]. Further, the peak intensity ratio (I_D_/I_G_) of N-GQDs was 0.85. This value points out that defects are introduced in the conjugated carbon backbone of N-GQDs via N-atom intercalation [69].

Figure 3a displays the UV–vis spectrum of N-GQDs and PIN. The PIN (Figure 3a inset) displays characteristic bands at 230 and 270 nm, corresponding to the π–π* and n–π* transitions between the stacked lattice structures and along with the conjugated backbone structures, respectively [70]. The UV–vis spectrum of N-GQDs has an absorption band at ∼270 nm referring to π–π* transition of sp^2^ graphitic domains [62]. Further, a characteristic absorption band at ∼330 nm corresponds to an n–π* transition was also observed, confirming the N-doping in graphene lattices. The aqueous suspensions of PIN and N-GQDs were photographed under visible light and at 365 nm and shown in the inset image of Figure 3a. On irradiation at 365 nm, the light yellow-colored N-GQDs solution exhibits a cyan fluorescence color. However, PIN has no characteristic PL. Further, the luminescent properties of N-GQDs were explored using PL measurements. The N-GQDs showed the strongest luminescence at 420 nm (Figure 3b). Upon excitation at 420 nm, N-GQDs displayed PLE peaks at ∼247 and 310 nm. It is interesting to observe that the PL emission of N-GQD is excitation wavelength dependent (Appendix A). Further, it can be concluded that the detected cyan blue PL emission is attributed to the irradiation decay of excited electrons from LUMO to HOMO. A detailed explanation is given in Appendix A.

### 3.2. The Electrochemical Response of Dopamine (DA) on Modified Electrodes

The CV responses of the DA electrochemical redox process (1 mM DA, 0.1 M PBS) at GCE, PIN/GCE, and N-GQDs/GCE are shown in Figure 4. It is evident from Figure 4 that the reduction/oxidation current of PIN/GCE and GCE is feeble, suggesting that these electrodes may not catalyze the oxidation of DA.

Further, the difference between the anodic and cathodic peaks (ΔE_p_) in the CV response of the N-GQD/GCE was calculated as 0.158 V, which is higher than the value of $\frac{59}{n}$ (n = 2). Thus, a quasi-reversible two-electron process under optimized conditions is verified [71], whereas ΔE_p_ in the CV response of the GCE was calculated as 0.05 V. Appendix A shows the LSV response of 1 mM DA (0.1 M PBS, pH 7.4) at GCE, PIN/GCE, and N-GQDs/GCE at a scan rate of 10 mV s^−1^ (accumulation time of 180 s). To demonstrate the adsorption ability of N-GQDs/GCE, we performed a pre-concentration experiment Appendix A. The detailed measurement procedure was presented in Appendix A. 

### 3.3. Mechanism of Electrochemical Oxidation of DA at N-GQDs/GCE

The mechanism of DA oxidation at N-GQDs/GCE was explained by employing a wider potential window of −0.7 to +0.9 V. Figure 1 shows two pairs of redox peaks within this electrochemical window. Thus, the mechanism of the electrochemical behavior of DA at N-GQDs/GCE can be represented as follows: The peak one comprises DA oxidation to dopamine-o-quinone (DOQ), which is a two-electron transfer process. Peak two appears through the reduction of DOQ to DA (reaction 1) [72]. Then, DOQ undergoes 1, 4-Michael addition, resulting in the formation of leucodopaminechrome (LC) (reaction 2). DA will be predominantly in the protonated form in a solution of pH below 8.9 since the pKa of DA is 8.9, and thereby the amino group of DA shows poor nucleophilicity towards the 1, 4-Michael addition reaction. Nevertheless, the unprotonated DOQ found in a solution of pH 7.4 undergoes a cyclization reaction, which manifests in peak three in the cyclic voltammogram. Consequently, reversible oxidation of LC to dopaminechrome (DC) occurs at pH 7.4 (reaction 3). In summary, the first electrochemical step (DA/DOQ) is defined by the redox couple that happened at the positive potentials, whereas the second electrochemical step (LC/DC) is manifested by the redox couple that happened at the negative potentials.

These two electrochemical steps are sandwiched by a homogeneous chemical reaction called the cyclization of DOQ [73]. Accordingly, the performance of DA at N-GQDs/GCE can be categorized as an ECE mechanism, in which E represents the electrochemical steps and C defines the chemical steps. Though the ECE mechanism explains the complete route in the DA oxidation, the DA/DOQ redox couple is used in the body of the work for the determination of I_pa_/I_pc_.

### 3.4. Optimization of Analytical Parameters

The electrochemical response of DA was evaluated by CV in 0.1 mM DA in various pH values (5.0–9). Appendix A shows that the best I_pa_ value was obtained at pH = 7.4. It is also noted that at higher pH values (pH > 7.4), the I_pa_ response decreases. Accordingly, electrochemical detection studies of dopamine were carried out in a buffer solution of pH 7.4. Appendix A represents the CV responses at different scanning rates (v). Appendix A portrays that the I_pa_/I_pc_ value approaches 1 in all scan rates and quasi-reversible redox kinetics of dopamine at N-GQDs/GCE is evident [40]. Furthermore, the plot of log (I_p_) versus log (v) gives a slope value of 0.8, denoting mixed adsorption and diffusion-controlled reaction mechanisms at the N-GQDs/GCE [74]. The dominance of the diffusion-controlled mechanism over adsorption is also validated in Appendix A, where I_p_ values linearly vary with the square root of the scan rate [40,75]. 

The following equation was used for the calculation of the number of electrons involved in the oxidation of DA at N-GQDs/GCE [71],
$$I_{p}=\frac{\mathrm{nFQ}\vartheta }{4\mathrm{RT}}$$
where I_p_ denotes the anodic peak current, T represents the temperature in Kelvin (298 K), Q defines the sum of charge integrated from the area of cyclic voltametric peak, R is the universal gas constant (8.314 J mol^−1^ K^−1^), n denotes the number of electrons transferred and F is the Faraday constant (96,500 C mol^−1^). Here, the calculated value of the number of electrons transferred (n) is 1.69 (≈2) in the electro-oxidation of DA. Therefore, the involvement of two protons and two electrons in the oxidation of DA can be validated. The LSV responds with the accumulation time at open circuit potential vs. Ag/AgCl in the range of 0–210 s at the N-GQDs/GCE and is shown in Appendix A. It is found that the I_p_ values increase with the increase in accumulation time up to the 180s, and thereafter remain almost constant (Appendix A). Hence, 180 s was selected as the best accumulation time in the determination of DA.

### 3.5. Mode of Electrochemical Sensing of DA at N-GQDs/GCE

The manner in which N-GQDs excel in DA sensing was depicted as follows. First of all, the interaction of N-GQDs with DA at the electrode/electrolyte interface was referred by the previous pre-concentration experiment. At pH 7.4, the positively charged DA molecule showed strong electrostatic attraction towards electronegative hydroxyl and carboxyl groups of N-GQDs. Likewise, the conjugated large π bonds in N-GQDs interact with DA via the π–π stacking force. Additionally, the positively charged DA might show cation–π interaction with conjugated large π bonds in the N-GQDS. As inferred from the characterization results, the higher proportions of pyrrolic-N and graphitic-N in N-GQDs can efficiently improve its sensing performance. Such graphitic-N can boost the catalytic performance of N-GQDs as it offers additional electron density and preserves the charge carrier mobility. Likewise, the presence of pyrrolic N in graphene lattices can produce strong disorders and defects, which will improve its adsorption properties [31].

In situ N-doping might modify the HOMO–LUMO energy gap (H–L gap) of the N-GQDs, which in turn alters the electron transfer barrier. Hence, the oxidation current (rate) will be altered. A small H–L gap (~1.02 eV) energetically favors the removal of electrons from HOMO and the addition of electrons to LUMO, thereby forming an activated complex. This denotes high electrocatalytic reactivity and low kinetic stability of the N-GQDs/GCE [76]. The cyclic voltammetry and Nyquist plot for Fe(CN)_6_^3−^ solution at the N-GQDs/GCE in comparison to the PIN/GCE validated the improved electrocatalytic performance of N-GQDs/GCE (Appendix A). The interpreted data are summarized in Appendix A.

The FT-IR spectra of DA, DA-adsorbed N-GQDs, and N-GQDs are displayed in Appendix A. It is evident from the spectra that the N-H/O-H stretching vibrations of N-GQDs were slightly shifted toward a higher wavenumber in DA-adsorbed N-GQDs, suggesting the hydrogen bond formation possibility between the N-H/OH groups and the DA hydroxyl groups. Similarly, a slight shift of C-NH-C stretching from 1380 to 1372 cm^−1^ in DA-adsorbed N-GQDs was also evident from Appendix A. Moreover, C=C stretching vibrations within the graphitic lattice were shifted to 1639 cm^−1^. These results confirm the cation–π or π–π stacking interactions between the oxidized state of DA and the conjugated large π bonds of N-GQDs.

### 3.6. Characteristics of N-GQDs/GCE Sensor

#### 3.6.1. Interference Investigation

The oxidation potentials of uric acid (UA), ascorbic acid (AA), and DA are close to each other and often coexist in several physiological fluids. Thus, it is important to confirm the selective sensing ability of the N-GQDs/GCE toward dopamine [77]. So, the possible interferences from AA and UA were investigated by CV and LSV responses. The LSV and CV response of 1 mM AA (trace a), 1 mM UA (trace b), 0.01 mM DA (trace c), DA in the presence of AA (100 fold, trace d), UA (100 fold, trace e), and mixtures of other interfering bioactive molecules and ions (trace f) are shown in Figure 5a and inset of Figure 5a. As known, UA and AA will be negatively charged, whereas DA is positively charged at pH 7.0 [78]. Consequently, the anionic groups of N-GQDs could attract positively charged DA, and the transfer of electrons via π–π superposition interaction between DA and N-GQDs was facilitated. However, the movement of UA and AA towards the N-GQDs/GCE surface were declined. Figure 5a clearly shows that UA and AA hardly weaken the anodic peak currents for DA. Further, the interference of bioactive molecules and ions such as Na^+^ (10b-a), K^+^ (10b-b), Ca^2+^ (10b-c), Zn^2+^ (10b-d), Mg^2+^ (10b-e), Co^2+^ (10b-f), glucose (10b-g) (100 fold), citric acid (10b-h), sucrose (10b-i), alanine (10b-j), phenylalanine (10b-k), glycine (10b-l), cysteine (10b-m), Fe^3+^ (10b-n), urea (10b-o) (50 fold), Al^3+^ (10b-p), valine (10b-q), ascorbic acid (10b-r), uric acid (10b-s) and dopamine (10b-t) (100 fold) was evaluated and was manifest as a to r in Figure 5b. Since these interfering substances could not affect the analytical performance of the N-GQDs/GCE, the high selectivity of this EC sensor was validated.

Further, we evaluated the amperometric response of the N-GQDs/GCE for 0.05 mM DA and different interfering species at an oxidation peak potential of +0.3 V. The results of the amperometric study are displayed in Figure 5c. The changes in the amperometric current (1.1 to 6.9 µA) were evident with the addition of 0.05 µM dopamine, while respective stepwise addition of interfering species (100 fold concentration) including urea, UA, AA, glucose, sucrose, cysteine, NaCl, KCl, MgCl_2_, valine, alanine, phenylalanine, MnCl_2_, and FeCl_3_ in 0.1 M PBS every 60 seconds hardly changed the current response in comparison to DA. Hence, the excellent selectivity of the EC sensor toward DA detection was proved. In addition to this, the CV response of each of these interfering molecules and ions is also taken and a plot of analytes vs. current I_pa_ is shown in Figure 5d.

#### 3.6.2. Stability, Reproducibility, and Repeatability

Stability, reproducibility, and repeatability studies were conducted to validate the applicability of N-GQDs/GCE as EC sensors. The reproducibility was studied using five different and freshly prepared N-GQDs/GCE in 1.0 mM DA (0.1 M PBS, pH 7.4), and the CV responses are shown in Appendix A. The anodic peak current showed a relative standard deviation (RSD) of 2.2%, suggesting the excellent consistency of electrode preparation, and that the electrode is reproducible. The repeatability of the fabricated N-GQDs/GCE was examined by performing a series of repetitive measurements using the same electrode in the presence of DA and low RSD values were obtained (Appendix A). The RSD of 2.7% is accepted (n = 10), indicating that the sensor has good repeatability. The cyclic stability performance of the N-GQDs/GCE was evaluated by CV responses in 1 mM DA in PBS (0.1 M). As shown in Appendix A, in the CV responses from the first cycle to the 50th cycle, only 1.08% of the peak current was reduced. Hence, the significant cyclic stability of N-GQDs/GCE for the sensing of DA molecules can be demonstrated. In addition, the N-GQDs/GCE also showed excellent storage stability. The N-GQDs/GCE in 0.1 M PBS was stored at room temperature and the CV response of the electrode in 1 mM DA solution over a single day was evaluated. As shown in Appendix A, the i_pa_ retains 97.12% of its original response over 17 days and this outcome validates the stable EC sensing performance of N-GQDs/GCE for DA detection.

#### 3.6.3. Standard Curves, Linear Response Ranges, and Limit of Detection

The CV responses for successive additions of DA (0.001–100 μM) in 0.1 M PBS (pH 7.4) at the N-GQDs/GCE were recorded in Figure 6a. As can be seen, the anodic peak current increased linearly with the increase in the DA concentration in the range of 0.001–100 μM. The calibration curve of anodic peak current values within 0.001–0.1 μM of DA was shown in Figure 6b. The corresponding linear regression equation is,
i_pa_ = 10.25C_DA_ + 5.857      R^2^ = 0.9980

From the figure, the detection limit (DL) of 0.2 nM, the sensitivity of 10.25 μA/μM, and the quantification limit (QL) of 0.65 nM were estimated.

The calibration curve of anodic peak current values of DA at the N-GQDs/GCE sensor in the linear range from 0.5 to 100 µM was shown in the inset of Figure 6b. The corresponding linear regression equation is,
i_pa_ = 0.1954C_DA_ + 9.761      R^2^ = 0.9917

LOD and LQD are calculated using the following equations,
$$LOD=3\;\frac{S}{b}\;$$
$$LQD=10\;\frac{S}{b}\;$$
where b corresponds to the slope of the calibration curve obtained from the CV and S denotes the standard deviation of the blank solution. 

The LSVs of DA (0.001–1000 μM) at N-GQDs/GCE under optimal accumulation conditions are shown in Figure 6c. The cathodic peak at 0.17 V, responding linearly to DA concentration was selected as our examination signal. 

The calibration curve of anodic peak current within the linear range of 0.001–0.5 μM was depicted in Figure 6d. The corresponding linear regression equation is,
i_pa_ = 15.95C_DA_ + 9.061      R^2^ = 0.9980

From the figure, the detection limit (DL) of 0.15 nM, the sensitivity of 15.95 μA/μM, and quantification limit (QL) of 0.5 nM were estimated.

The calibration curve of anodic peak current values within the linear range of 0.5–1000 μM was shown in the inset of Figure 6d. The corresponding linear regression equation is,
i_pa_ = 0.0268C_DA_ + 18.81      R^2^ = 0.9654

The comparison of the sensing of DA at N-GQDs/GCE with previous reports was illustrated in Table 1. It was evidenced that reasonable sensitivity, linear concentration range, and lower detection limit were the successes of the proposed N-GQDs/GCE.

### 3.7. Analytical Performance

#### Real Urine Sample Analysis

The practical applicability of N-GQDs as an EC sensor in real samples was evaluated by studying its analytical performance in real urine samples (Appendix A). Various concentrations of DA were mixed with the urine sample, and the corresponding LSV and CV response of DA (5–25 μM) at the N-GQDs/GCE sensor are shown in Appendix A. The recovery range of the proposed sensor was from 98.4 to 102.29% and an error % of 0.8–2.9% which demonstrated that other interferences could not work. Moreover, the obtained results established a good agreement with other reported methods [54,79,80]. 

## 4. Conclusions

In our work, we report a single-step hydrothermal strategy to synthesize N-GQDs from polyindole (PIN), and these N-GQDs were employed as an electrochemical sensor for dopamine (DA) detection. Various characterization results were employed to confirm the successful formation of N-GQDs. The effective N-doping in graphene lattices might result in exciting properties such as enhanced conductivity, improved electron transfer capability, and a low H–L gap, which aids in unique EC sensing behavior towards dopamine. The selective interaction of N-GQDs with the dopamine through electrostatic H-bonding, π–π interactions, and ring stacking owing to the base plane aromatic N-hetero cycles and O-rich hydrophilic side functionalities possibly account for the excellent selectivity of the EC sensor. Remarkably, it could detect as low as 0.15 nM of DA with a linear range of 0.0011000 µM, showing excellent sensitivity and could efficiently distinguish between uric acid, ascorbic acid, and other interfering molecules and ions. In addition, the N-GQDs/GCE displayed better reproducibility, stability, and reusability. Furthermore, the EC sensor possessed remarkable dopamine sensing in real samples. 

## Data Availability

Data is contained within the article or Appendix A.

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
