# Peer review of "Polyindole-Derived Nitrogen-Doped Graphene Quantum Dots-Based Electrochemical Sensor for Dopamine Detection"

_biosensors, 2022, doi:10.3390/bios12121063_

Round 1

Reviewer 1 Report

This work reports the application of a nitrogen-doped graphene quantum dots-modified electrode for the determination of dopamine in real urine samples. The N-GQDs were synthesized from polyindole (PIN) via a hydrothermal synthetic strategy. The modified electrode was prepared by drop-casting of an aqueous suspension of N-GQDs on a glassy carbon electrode. The work is quite interesting and rich in data. However, there are still some doubts to be clarified. The fact that N-GQDs are soluble in water (and therefore soluble in the supporting electrolyte) needs to be clarified. How they would remain stable in the form of a film on the glassy carbon electrode? Therefore, I recommend that a minor revision be prepared so that the manuscript can be published in Biosensors.

General comments:

1. “The presence of carbonyl groups on the surface which make N-GQDs highly water-soluble (Line 206)”. Thus, as the authors guarantee that the N-GQDs formed a stable film on the surface of the GCE. This needs to be clarified.

2. The authors, like many others, confuse the terms "detection" and "determination". Detection is qualitative by nature, while determination always is quantitative. Qualitative analysis is the detection of the presence of ions or compounds in an unknown sample, for example. The term "determination" refers to quantitative analysis to obtain data on the amount of analyte by weight or by concentration of an element or a compound in a sample. Therefore, most of the words “detection" in the manuscript should be replaced by the term "determination" (or "quantitation" or "assay") if quantitative assays are involved.

3. Update the terminologies for the electrochemical methods according to new IUPAC recommendations. See the information in Pure and Applied Chemistry, 92 (2020) 641–694. Current is I.

4. Use the acronyms GCE, PIN/GCE and N-GQDs/GCE to refer to the electrodes.

5. Supplemental Material is confusing. There are repeated sentences throughout the Supplementary Material text. Please review.

Specific comments:

1. Introduction:

a. Lines 62-65. We commonly describe the negative features of chromatographic techniques to laud our (electroanalytical) technique. I believe that all analytical techniques have advantages and disadvantages and that they all have space and function in scientific research. Thus, we can describe the positive characteristics of electrochemical methods, without diminishing the other techniques. This is just an opinion. Authors do not need to answer or comment.

b. Lines 65-67. One of the great advantages of electrochemical methods is the portability of the equipment. This is a great advantage over chromatographic methods. Add "accuracy" and "portability of equipment" to the list of characteristics of electroanalytical methods and sensors. Add the Chemosensors, 10 (2022) 357 reference to validate this information.

2. Materials and methods:

a. Line 109-110. Do not use the acronym PBS for "phosphate-buffered solution". PBS is the accepted acronym for "phosphate-buffered saline" which contains 0.9% NaCl to warrant physiological ionic strength. See the Sigma Aldrich catalog (product P5368), for example. Also, see: https://en.wikipedia.org/wiki/Saline (medicine). Unfortunately, PBS is often wrongly used as an acronym for "phosphate buffer(ed) solution" in the literature but this is wrong and can cause confusion.  Does the buffer employed by the authors really contain 0.9% NaCl (or other electrolytes such as MgCl2)? If yes, please specify. See: https://en.wikipedia.org/wiki/Phosphate-buffered_saline.

b. Add the characterizations section after the synthesis sections.

c. Line 125. The correct is mL, not ml.

d. Line 140. Information on the ultracentrifugation process must be provided. how many rpm? How many minutes? How many times was the procedure performed?

e. Create a section on sample preparation. Inform the city and country where the samples were obtained.

f. Standardize the description of the equipment: model (company, country).

3. Results and discussion:

a. Line 242. The authors claim that N-GQDs electrocatalyze the dopamine oxidation/reduction reaction. For such a claim, it is necessary to discuss the potential shifts of dopamine in relation to naked GCE and PIN/GCE.

b. Line 262. The figure caption is very poor in information. Enter more experimental information.

c. The Scheme S3 must be transferred to the manuscript.

d. Lines 284-286. A linear relationship in the Ip vs. v1/2 indicates that the electrochemical process was diffusion controlled. For this reason, when logarithm is applied to the graph, we say that a slope close to 0.5 indicates diffusion. The authors need to review this confusing sentence. For more information on the equations, derivatives and graphs of this study, see Talanta, 62 (2004) 912–917.

e. Line 368-370. Reproducibility is the value below which the absolute difference between two single test results obtained by the same method on investigated test compounds under different conditions such as different analysts, different equipment, and different laboratories. For more information, see The Open Analytical Chemistry Journal, 5 (2011) 1–21. The authors performed repeatability tests.

f. Table 1. Recent work (2021-2022) has been reported for the determination of dopamine. The Table must contain recent work. Some suggestions to add to the table: Nanocomposites, 8 (2022) 155-166 and Biochemical Engineering Journal, 186 (2022) 108565. 

g. Lines 438-439. “Moreover, the obtained results established a good agreement with other reported methods”. What other methods? Cite the work reference.

5. Supplementary Material:

a. Page 9. Laviron's equation (Ep. vs. log v plot) is for irreversible systems only.

b. Page 9. “The estimated value of ks is 3.01 s−1, which is significantly higher than that reported earlier.” Compared to what value? Cite reference. Or do the calculation using GCE only.

c. What is the geometric area and electroactive area of the bare GCE? Provide the values.

Reviewer 2 Report

Manuscript ID: biosensors-2017573

Anjitha et al reported the “Polyindole-Derived Nitrogen-Doped Graphene Quantum Dots Based Electrochemical Sensor for Dopamine Detection”. This work is interesting and suitable for publication in Biosensors. However, the authors should address the following comments before publication. Hence, I recommend a major revision.

1.     Previously numerous works were reported based on the N-GQDs using the electrochemical sensor for various analyte molecule detection. The author should compare the articles with your work in the introduction section. 

2.     The author should give the proper citation for the introduction of PIN.

3.     The author mentioned “3 g indole in 50 ml of the ethanol-water mixture”. What about the ratio of the ethanol-water mixture?

4.     The author claimed “Figure 1a depicts the uniform size distribution” but the sizes are not uniform. How to explain this?

5.     Explain the particle distribution diagram and SAED pattern result in section 3.1.

6.     The author should carefully check the superscript and subscript of the units in your manuscript. 

7.     In Line No: 217, change the acronym Uv-Vis to UV-Vis. Check the acronyms thoroughly in the manuscript.

8.     Generally, the bare GCE shows a poor electrochemical response toward the detection of DA. But Figure 4(a), no response of bare GCE for DA oxidation. The author should check again the DA response at bare GCE.

9.     The author should check Figure S5a in the mentioned pH values. Why did the author choose voltammetry techniques (CV and LSV) to be used for the detection of DA? What are the advantages of voltammetry techniques over (i-t) Amperometry?

10.  Line No: 416, “The cathodic peak at 0.36 V” please mention correctly in the revised manuscript.  

Reviewer 3 Report

The presented work is highly relevant, performed at a high methodological level, the conclusions are reasonable and clear. It is only necessary to correct the flaws in the quality of the text representation (a small mess with missing or extra spaces) + Figure 5 looks like it is placed in the gap of the main text, which is confusing when reading (Page 10 line 381). With the exception of these moments, the manuscript makes a completely positive impression

Round 2

Reviewer 2 Report

The authors have explained the peer review comments accordingly and improved the manuscript. The quality of this article has been improved significantly and is suitable for publication.